# Searching for Systems with Planar Hexacoordinate Carbons

Diego Inostroza [1,2], Luis Leyva-Parra [1,2], Osvaldo Yañez [3,*], José Solar-Encinas [1,2],
Alejandro Vásquez-Espinal [4], Maria Luisa Valenzuela [5] and William Tiznado [1,*]

1    Computational and Theoretical Chemistry Group, Departamento de Ciencias Químicas, Facultad de Ciencias Exactas, Universidad Andrés Bello, Av. República 275, Santiago 8370146, Chile; dinostro92@gmail.com (D.I.); l.leyvaparra@uandresbello.edu (L.L.-P.)
2    Programa de Doctorado en Fisicoquímica Molecular, Facultad de Ciencias Exactas, Universidad Andrés Bello, Av. República 275, Santiago 8370146, Chile
3    Núcleo de Investigación en Data Science, Facultad de Ingeniería y Negocios, Universidad de las Américas, Santiago 7500000, Chile
4    Química y Farmacia, Facultad de Ciencias de la Salud, Universidad Arturo Prat, Casilla 121, Iquique 1100000, Chile; alvasquez@unap.cl
5    Grupo de Investigación en Energía y Procesos Sustentables, Instituto de Ciencias Químicas Aplicadas, Facultad de Ingeniería, Universidad Autónoma de Chile, Av. El Llano Subercaseaux 2801, Santiago 8900000, Chile; maria.valenzuela@uautonoma.cl
*    Correspondence: oyanez@udla.cl (O.Y.); wtiznado@unab.cl (W.T.)

**Abstract:** Here, we present evidence that the $D_{2h}$ $M_2C_5^{0/2+}$ (M = Li-K, Be-Ca, Al-In, and Zn) species comprises planar hexacoordinate carbon (phC) structures that exhibit four covalent and two electrostatic interactions. These findings have been made possible using evolutionary methods for exploring the potential energy surface (AUTOMATON program) and the Interacting Quantum Atoms (IQA) methodology, which support the observed bonding interactions. It is worth noting, however, that these structures are not the global minimum. Nonetheless, incorporating two cyclopentadienyl anion ligands (Cp) into the $CaC_5^{2+}$ system has enhanced the relative stability of the phC isomer. Moreover, cycloparaphenylene ([8]CPP) provides system protection and kinetic stability. These results indicate that using appropriate ligands presents a promising approach for expanding the chemistry of phC species.

**Keywords:** planar hexacoordinate carbon; global minima; kinetic stability; DFT computations; chemical bonding analysis



## 1. Introduction

Chemists are fascinated by new chemical entities with exotic, non-classical structures, so they seek to rationalize these new systems based on known rules (concepts) or propose exceptions, and new methods, to achieve this goal. Molecules with planar hypercoordinate carbon atoms, which violate the well-established rule of Van 't Hoff and Le Bel (regarding tetrahedral four-coordinate carbon), are particularly puzzling. Although these species, in the beginning, were considered experimentally inaccessible, in 1968, Monkhorst evaluated *in silicon* methane stereomutation through a planar tetracoordinate carbon (ptC) transition state [1]. Subsequently, in 1970, Hoffmann and co-workers proposed different approaches to stabilize a ptC to achieve a thermally accessible transition state for a racemization process [2]. These studies inspired other chemists who finally allowed the identification of viable ptC compounds [3–7].

In the last 50 years many ptC compounds have been reported, some synthesized or identified in the gas phase, and others designed *in silicon* [3–5,8–11]. In more recent years, the chemistry of the family has been extended to species in which the carbon coordination number is greater than four (penta [12–21] and hexacoordinated [22–25]). The examples for each group decrease according to the coordination, with only 15 structures reported as global minima with a hexacoordinated carbon in the plane.

In 2004, Merino et al. reported that their exploration of the potential energy surface of $C_5^{2-}$ revealed the existence of a local minimum with a planar tetracoordinate carbon for this dianionic cluster [26,27]. However, the ptC $C_4^{2-}$ isomer is 48.4 kcal·mol$^{-1}$ above the global minimum (GM), which is an angular $C_{2v}$ structure (at the B3LYP/6-311++G(2d) level). In addition, these authors also found that adding metal cations (M$^{n+}$) to the $C_5^{2-}$ structure generates species with the targeted planar tetracoordinate carbon, which is stabilized exclusively by electronic factors. They tested cations from groups 1, 2, 11, and 13 of the periodic table. They concluded that the lithium salt $C_5Li_2$ is the most plausible candidate for experimental detection based on its energetic profile of the isomerization reaction. To our knowledge, none of these species have been detected experimentally. However, this study has inspired the identification of other viable gas-phase ptC systems (global minima).

In this work, we have raised some questions regarding the systems mentioned in the previous paragraph (results and discussion, Section 3). The first question is related to the proper coordination of the putative ptC (Section 3.1); since the natural population analysis (NPA) charge on the ptC is negative and that of the counterions is positive, and the ptC-M distance is smaller than the sum of the Van der Waals radii, an attractive ptC-M electrostatic interaction is to be expected. Therefore, this would genuinely be a planar hexacoordinated carbon (phC). The second question is related to the viability of any of these species. For this, we have explored the PES of some candidates (Section 3.2), showing that they do not correspond to global minima. In addition, we have explored the alternative of using ligands (cyclopentadienyl anion and 8-ring cycloparaphenylene) to provide thermodynamic and kinetic stability to the phC system (Section 3.3). Our results reveal that these systems are indeed phCs, with four covalent and two ionic interactions. Additionally, ligands provide relative energetic and kinetic stability to some systems, especially $Ca_2C_5^{2+}$. Therefore, we expect this study to open a pathway in the design of phCs with synthetic feasibility.

## 2. Computational Details

The PES of the $Li_2C_5$ system was explored using the AUTOMATON program [28,29]; in the search process, optimizations were performed at the PBE0 [30] /SDDAll [31–35] level. Then, the lowest energy minima were reoptimized at the ωb97XD [36] /def2-TZVP [37] level (both in singlet and triplet states). For the other systems $C_5M_2$ (M = Na, K, Al-In) and $C_5M_2^{+2}$ (Be-Ca, Zn), the lowest energy minima obtained for $C_5Li_2$, were used as starting structures, then optimized at the PBE0/SDDAll level and reoptimized at the ωb97XD/def2-TZVP level, the latter being the level considered for the discussion of relative energies. The lowest energy isomers confined between aromatic ligands, benzene ($C_6H_6$) and cyclopentadienyl anion ($C_5H_5^-$) were also optimized for the neutral and dicationic cases, respectively. Finally, the phC isomer of $Ca_2C_5^{2+}$ was optimized inside an 8-ring cycloparaphenylene, where two $C_5H_5^-$ have substituted two benzenes. The vibrational frequencies were checked to verify the structures as true minima on the PES. DFT computations were performed with Gaussian 16 software (Rev. B.01) [38]. The dynamic behavior of $[(C_5H_5^-)]_2[Ca_2C_5]^{2+}$ and $[8\text{-CPP}]^{2-}[Ca_2C_5]^{2+}$ were assessed through Born Oppenheimer molecular dynamics (BOMD) simulations [39] on Gaussian 16 software at the PBE0-D3/SDDAll level. The BOMD simulations involve 20 ps at 500 K and within the NVT ensemble.

To gain insights about chemical bonding, we used different methods: Wiberg bond indices (WBI) [40], natural population analysis (NPA) [41], and the adaptive natural density partitioning (AdNDP) method [42,43]. These approaches are based on the natural bonding orbital (NBO) method and were performed at the ωb97XD/Def2-TZVP level. The WBI and NPA were computed with the NBO 6.0 code [44], and the AdNDP analysis was performed using Multiwfn 3.8 [45]. The molecular structure and AdNDP results were visualized using CYLview 2.0 [46] and VMD 1.9.3 [47], respectively.

## 3. Results and Discussion

### 3.1. Defining Whether the Systems Are ptC or phC

Table 1 shows the NPA charges, WBI values, and bond distances for the $D_{2h}$ structure for the combinations considered in this study. The charges are negative in carbons ranging from $-0.35$ to $-0.23$ |e| for the hypercoordinate one. The WBI values indicate a strong covalent connection between the C atoms and no or very weak C m covalent interaction, especially of the peripheral Cs, in agreement with its smaller C-C and larger M-C distances. Given that the charges on the hypercoordinate C and the metals have an opposite sign, one may inquire about a meaningful attractive electrostatic interaction between them that would contribute to the stability of this structure. Moreover, if attractive, these interactions would qualify them as phC species. The M-C distances are slightly greater than those of a single bond, according to the Pyykkö covalent radii [48]. However, they are much shorter than the sum of their van der Waals radii [49] (see Table 1). Therefore, it seems that, in these structures, the C is hexacoordinate.

**Table 1.** Bond lengths (r, Å), natural charges (q, |e|) and Wiberg bond indices (WBI) of the $M_2C_5^{0/2+}$ (M = Li-K, Be-Ca, Al-In and Zn) computed at the $\omega b97XD/Def2-TZVP$ level [a].

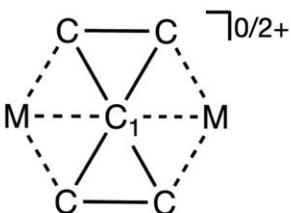

| System | $q(C_1)$ | $q(C)$ | $q(M)$ | $r_{C_1-C}$ | $r_{C_1-M}$ | $r_{C-M}$ | $r_{C-C}$ | $WBI_{C_1-C}$ | $WBI_{C_1-M}$ | $WBI_{C-M}$ | $WBI_{C-C}$ |
|---|---|---|---|---|---|---|---|---|---|---|---|
| $Li_2C_5$ | $-0.27$ | $-0.36$ | $0.87$ | $1.49$ | $2.20$ | $2.04$ | $1.32$ | $0.98$ | $0.01$ | $0.10$ | $1.70$ |
| $Na_2C_5$ | $-0.25$ | $-0.38$ | $0.88$ | $1.49$ | $2.61$ | $2.36$ | $1.32$ | $0.99$ | $0.01$ | $0.09$ | $1.71$ |
| $K_2C_5$ | $-0.24$ | $-0.39$ | $0.91$ | $1.50$ | $2.98$ | $2.68$ | $1.32$ | $0.99$ | $0.01$ | $0.07$ | $1.71$ |
| $Be_2C_5^{2+}$ | $-0.35$ | $-0.25$ | $1.67$ | $1.41$ | $1.98$ | $1.81$ | $1.32$ | $0.97$ | $0.01$ | $0.26$ | $1.65$ |
| $Mg_2C_5^{2+}$ | $-0.29$ | $-0.32$ | $1.79$ | $1.46$ | $2.37$ | $2.15$ | $1.32$ | $0.98$ | $0.01$ | $0.17$ | $1.69$ |
| $Ca_2C_5^{2+}$ | $-0.27$ | $-0.35$ | $1.84$ | $1.49$ | $2.64$ | $2.39$ | $1.32$ | $0.98$ | $0.02$ | $0.12$ | $1.70$ |
| $Zn_2C_5^{2+}$ | $-0.28$ | $-0.20$ | $1.55$ | $1.42$ | $2.35$ | $2.11$ | $1.32$ | $0.97$ | $0.02$ | $0.34$ | $1.69$ |
| $Al_2C_5$ | $-0.25$ | $-0.30$ | $0.74$ | $1.48$ | $2.52$ | $2.28$ | $1.33$ | $0.97$ | $0.02$ | $0.23$ | $1.67$ |
| $Ga_2C_5$ | $-0.23$ | $-0.30$ | $0.73$ | $1.48$ | $2.61$ | $2.35$ | $1.33$ | $0.98$ | $0.02$ | $0.24$ | $1.67$ |
| $In_2C_5$ | $-0.23$ | $-0.32$ | $0.75$ | $1.48$ | $2.82$ | $2.53$ | $1.33$ | $0.98$ | $0.01$ | $0.22$ | $1.68$ |

[a] The sums of Pyykkö's single-bond radii for the C–Li, C–Na, C–K, C–Be, C–Mg, C–Ca, C–Zn, C–Al, C–Ga, and C–In bonds are 2.08, 2.30, 2.71, 1.77, 2.14, 2.46, 1.93, 2.01, 1.99, and 2.17 Å, respectively. The sums of van der Waals radii for the C–Li, C–Na, C–K, C–Be, C–Mg, C–Ca, C–Zn, C–Al, C–Ga, and C–In bonds are 3.89, 4.27, 4.50, 3.75, 4.28, 4.39, 4.16, 4.02, 4.09, and 4.20 Å, respectively.

All the above background points to the central carbon's ionic bonding interactions with the M counterions. Hence, using an appropriate methodology to describe ionic interactions is mandatory. The IQA methodology is an accurate and chemically intuitive energy partitioning scheme that allows the analysis of chemical bonding in terms of covalent or ionic interactions [50–53]. This methodology has been used recently to gain insights into the bonding of the $NaBH_3^-$ system [54], a system that has been a challenge for the theoretical community dedicated to the analysis of chemical bonding [55–60]. IQA was fundamental to introducing collective interactions in organometallic systems [61]. It also supports the hexacoordination of C in the global minima $M_3CE_3^+$ (M = Li-Cs and E = S-Te) [24], with three covalent C-E and three ionic M-C interactions, becoming an indispensable method to rationalize coordination where there is a mixture of ionic and covalent interactions [21,62].

What is the interpretation of the chemical bond, according to IQA, of $D_{2h}$ $M_2C_5^{0/2+}$ (M = Li-K, Be-Ca, Al-In, and Zn) species? As shown in Table 2, the most important stabilizing interactions are between C-C bonds, mainly covalent ($V_{IQA}^{int}$: $-283.2$ to $-301.0$ kcal·mol$^{-1}$)

with a relatively weaker electrostatic destabilizing C-C interaction ($V_C^{int}$: 50.0 to 54.2 kcal·mol$^{-1}$). Additionally, the delocalization index ($\delta$) predicts C-C bond orders like those indicated by the WBI values. Therefore, IQA supports the coordination of the peripheral carbons with the central carbon by significant covalent C-C interactions. How about the interaction between the metallic counterions and the central C? IQA reveals that there is a stabilizing interaction (bonding), primarily electrostatic, and the $V_{IQA}^{int}$ ranges from $-36.3$ (M = K) to $-175.4$ kcal·mol$^{-1}$ (M = Be). Therefore, IQA supports the hexacoordination of C in all these structures, four C-phC (covalent) and two M-phC (ionic).

**Table 2.** Energy components (in kcal·mol$^{-1}$) of IQA for the $M_2C_5^{0/2+}$ (M = Li-K, Be-Ca, Al-In, and Zn) computed at the ωb97XD/Def2-TZVP level. $V_{IQA}^{int}$, $V_C^{int}$, and $V_{XC}^{int}$ are interatomic IQA interaction energy and their coulombic and exchange-correlation energy components, respectively. $\Delta E_{IQA}$ is the total integration error in IQA energies, and the delocalization indices are represented by $\delta$.

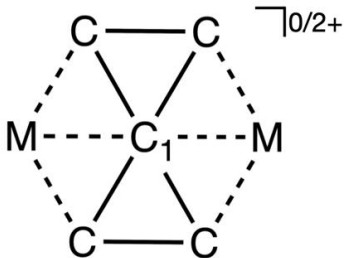

| System | $Li_2C_5$ | $Na_2C_5$ | $K_2C_5$ | $Be_2C_5^{2+}$ | $Mg_2C_5^{2+}$ | $Ca_2C_5^{2+}$ | $Zn_2C_5^{2+}$ | $Al_2C_5$ | $Ga_2C_5$ | $In_2C_5$ |
|---|---|---|---|---|---|---|---|---|---|---|
| $\Delta E_{IQA}$ | 0.0 | 0.0 | 0.0 | 0.0 | 0.0 | 0.0 | 0.0 | 0.0 | 0.0 | 0.0 |
| $V_{IQA}^{int}(C_1 - M)$ | −58.2 | −42.7 | −36.3 | −175.4 | −115.1 | −90.6 | −87.3 | −79.3 | −58.1 | −50.0 |
| $V_C^{int}(C_1 - M)$ | −54.6 | −39.8 | −33.0 | −169.7 | −110.3 | −82.5 | −73.6 | −73.3 | −49.1 | −42.8 |
| $V_{XC}^{int}(C_1 - M)$ | −3.7 | −2.9 | −3.3 | −5.7 | −4.8 | −8.2 | −13.7 | −6.0 | −9.0 | −7.2 |
| $V_{IQA}^{int}(C_1 - C)$ | −160.9 | −161.4 | −160.3 | −186.4 | −169.7 | −163.4 | −183.7 | −162.4 | −166.8 | −166.9 |
| $V_C^{int}(C_1 - C)$ | 24.3 | 23.0 | 22.7 | 21.0 | 22.7 | 20.7 | 10.0 | 26.2 | 19.3 | 19.1 |
| $V_{XC}^{int}(C_1 - C)$ | −185.2 | −184.4 | −182.9 | −207.4 | −192.4 | −184.0 | −193.7 | −188.6 | −186.0 | −186.0 |
| $V_{IQA}^{int}(C - M)$ | −82.0 | −73.5 | −70.4 | −205.6 | −157.4 | −130.3 | −112.5 | −137.1 | −94.2 | −86.8 |
| $V_C^{int}(C - M)$ | −71.1 | −59.5 | −52.7 | −181.3 | −133.6 | −98.2 | −43.9 | −102.3 | −50.9 | −46.5 |
| $V_{XC}^{int}(C - M)$ | −10.9 | −14.0 | −17.6 | −24.3 | −23.8 | −32.1 | −68.5 | −34.7 | −43.4 | −40.3 |
| $V_{IQA}^{int}(C - C)$ | −247.1 | −246.6 | −246.8 | −230.1 | −242.8 | −246.6 | −236.1 | −237.4 | −237.6 | −239.1 |
| $V_C^{int}(C - C)$ | 53.4 | 54.0 | 54.2 | 53.1 | 51.0 | 50.1 | 53.4 | 51.6 | 50.0 | 50.7 |
| $V_{XC}^{int}(C - C)$ | −300.5 | −300.7 | −301.0 | −283.2 | −293.7 | −296.7 | −289.6 | −289.0 | −287.6 | −289.8 |
| $\delta(C_1 - M)$ | 0.0 | 0.0 | 0.0 | 0.0 | 0.0 | 0.1 | 0.1 | 0.1 | 0.1 | 0.1 |
| $\delta(C_1 - C)$ | 1.1 | 1.1 | 1.1 | 1.1 | 1.1 | 1.1 | 1.1 | 1.1 | 1.1 | 1.1 |
| $\delta(C - M)$ | 0.1 | 0.1 | 0.2 | 0.2 | 0.2 | 0.3 | 0.5 | 0.3 | 0.4 | 0.4 |
| $\delta(C - C)$ | 1.7 | 1.7 | 1.7 | 1.6 | 1.7 | 1.7 | 1.6 | 1.6 | 1.6 | 1.6 |

We have also analyzed the bonding with the AdNDP method (see Figures 1 and S1–S3). It predicts that the covalent bonds are distributed on the $C_5$ fragment, with four lone pairs (one on each peripheral C), six single C-C bonds, two linking the peripheral Cs in pairs, and four linking these with the central C. Finally, AdNDP also recovers a delocalized π-bond throughout the $C_5$ fragment, confirming that these systems consist of the $C_5^{2-}$ dianion interacting with the metal counterions. AdNDP also retrieves the non-bonding electron pairs on the Al, Ga, In, and Zn counterions (Figures S1–S3).

### 3.2. Are phC Systems Global Minima Structures?

To answer this question, we have explored the potential energy surface of the combinations of interest using the constraints specified in the computational details. The putative global minima and other relevant minima are shown in Figure 2. It should be noted that the lowest energy structure for neutral systems is in a singlet state, not for ionic systems where the $D_{\infty h}$ triplet is more stable. However, the phC is only minimal in the singlet state. As can be seen, the phC structure does not correspond to the putative global minimum in any of the cases, being the closest isomer to the putative global minimum in the $M_2C_5$ com-

binations (M = Li-K; Al-In). The $Li_2C_5$ and $Na_2C_5$ phC structures have the closest energy to their corresponding putative global minimum, located above 9.0 and 11.4 kcal·mol$^{-1}$, respectively. This agrees with previous claims that the $D_{2h}$ $Li_2C_5$ cluster would be the most viable for experimental identification [63].

|  | E-LPs | E-E $\sigma$-bonds | E-E $\pi$-bond |
|---|---|---|---|
|  | 4 x 1c-2e | 6 x 2c-2e | 1 x 5c-2e |
| **E = C, M = Li** | ON = 1.89 \|e\| | ON = 1.97 − 1.78 \|e\| | ON = 1.99 \|e\| |
| **E = C, M = Na** | ON = 1.89 \|e\| | ON = 1.97 − 1.78 \|e\| | ON = 1.99 \|e\| |
| **E = C, M = K** | ON = 1.90 \|e\| | ON = 1.97 − 1.78 \|e\| | ON = 1.99 \|e\| |

**Figure 1.** Adaptive Natural Density Partitioning bonding pattern of the $M_2C_5$ (M=Li, Na, K) at the $\omega$b97XD/Def2-TZVP level.

### 3.3. Exploring Strategies to Stabilize phC Systems

Using ligands to protect is a common technique to provide stability and viability to atomic clusters [19,64,65]. Thus, computational studies on clusters that include protection models have also been increasing, including some clusters with hypercoordinate atoms [66–68]. However, these studies do not analyze the relative stability of the different lower-energy isomers confined within these protective ligands. Here, we have evaluated how the relative stability of the sandwich complexes of the more stable isomers changes. We have employed as ligands the aromatic rings benzene ($C_6H_6$) and cyclopentadienyl anion ($C_5H_5{}^-$). The former forms a sandwich with the neutral species, and the latter with the dicationic species, to have neutral complexes in all cases. The results are reported in Figures S4–S14. The use of ligands affects the relative energies; however, in no case is the phC structure preferred. The most striking case is the $[(C_5H_5{}^-)]_2[Ca_2C_5]^{2+}$ system, where the complex containing the phC (Figure 3) is now within 8.0 kcal·mol$^{-1}$ of the one that includes the putative global minimum. The barrier is reduced by half (relative to the isolated clusters, Figure 2) because of the $C_5H_5{}^-$ effect. This indicates that ligands should be explored in their protective role and in providing thermodynamic stability to the systems, as well.

Finally, we have studied the kinetic stability of the $[(C_5H_5{}^-)]_2[Ca_2C_5]^{2+}$ complex (Figure 3A) to explore viable phC species, knowing that the isolated phC cluster is a high-energy isomer on the PES but the $[(C_5H_5{}^-)]_2[Ca_2C_5]^{2+}$ is the closest energy isomer. Additionally, to provide more rigidity to the host ligand, we have placed the $C_5H_5{}^-$ rings inside a nanobelt (the 8-ring cycloparaphenylene, [8]CPP) to evaluate the dynamics of the $[8\text{-CPP}]^{2-}[Ca_2C_5]^{2+}$ complex (Figure 3B). Figure 4 shows results from ab initio BOMD simulations at 500 K for 20 ps. The structural fluctuations are plotted by the root mean square deviations curve (RMSD, in Å) versus simulation time. As Figure 4 shows, the RMSD has sharp changes, but the phC structure persists throughout the simulation. Additionally, two short movies extracted from the BOMD simulations are presented in the Supporting Information. Interestingly, structural fluctuations are approximately halved when using

the [8-CPP]$^{2-}$ ligand (median RMSD ≈ 0.8 Å) relative to free $C_5H_5^-$ ligands (median RMSD ≈ 1.5 Å).

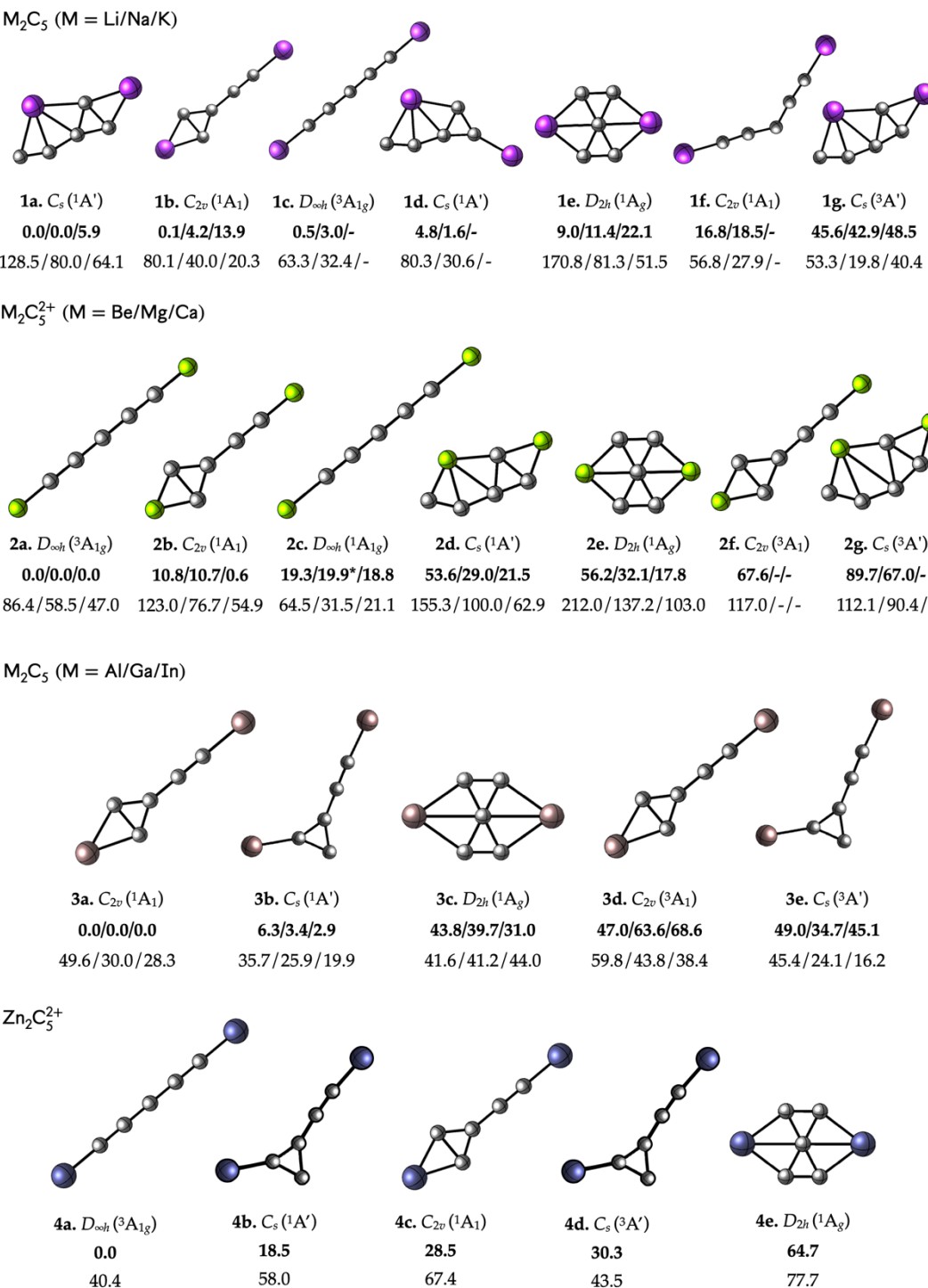

**Figure 2.** Putative global minimum and low-lying isomers of $M_2C_5^{0/2+}$ (M = Li-K, Be-Ca, Al-In, and Zn) clusters with their point group symmetries. Relative energies, including zero-point energy (ZPE) corrections in kcal·mol$^{-1}$ (**in bold**) and their lowest harmonic vibrational frequency in cm$^{-1}$ (below) at ωb97XD/Def2-TZVP level, are also shown. Cartesian coordinates of phC structures are depicted in Table S1. * The 2c structure for M = Mg is slightly bent in the center; therefore, it has a $C_{2v}$ symmetry.

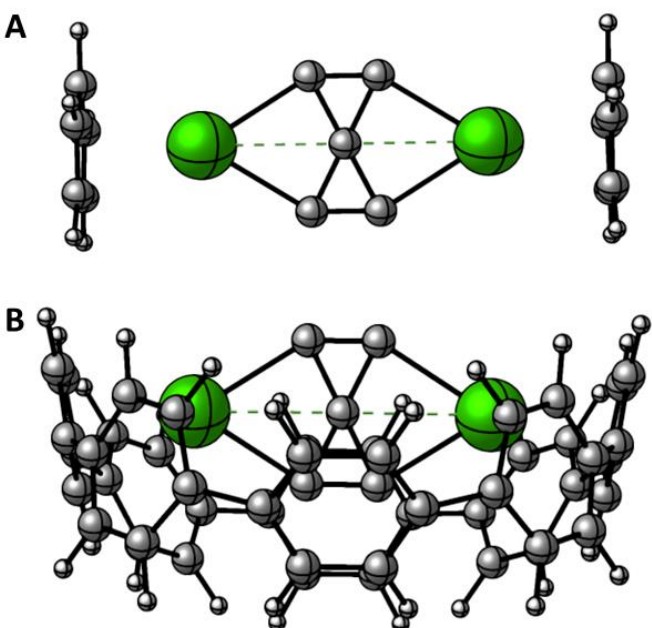

**Figure 3.** Low-lying isomers of (**A**) $[(C_5H_5{}^-)]_2[Ca_2C_5]^{2+}$ and (**B**) $[8\text{-CPP}]^{2-}[Ca_2C_5]^{2+}$.

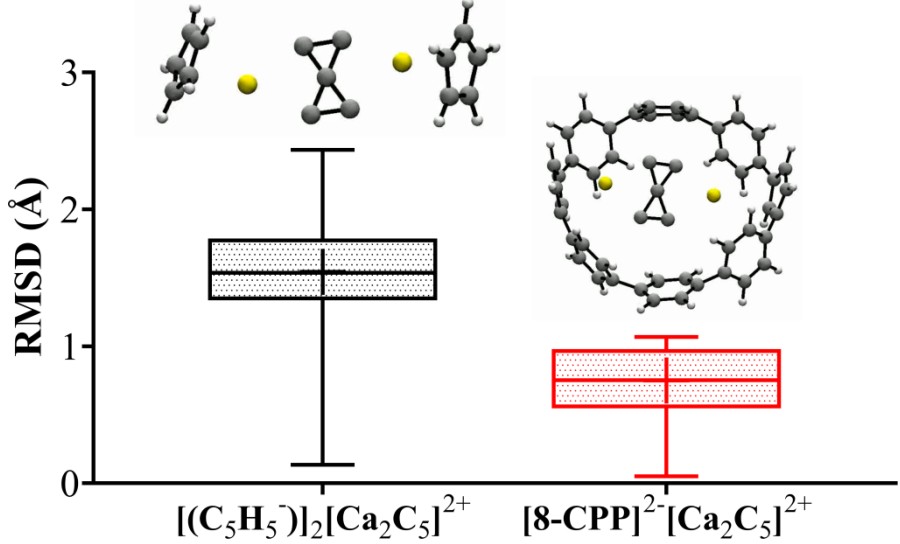

**Figure 4.** Boxplot of RMSD fluctuation of ab initio BOMD simulations for $[(C_5H_5{}^-)]_2[Ca_2C_5]^{2+}$ and $[8\text{-CPP}]^{2-}[Ca_2C_5]^{2+}$ systems.

## 4. Conclusions

It is shown that the $D_{2h}$ $M_2C_5{}^{0/2+}$ (M = Li-K, Be-Ca, Al-In, and Zn) clusters are planar hexacoordinate carbon species with four covalent and two electrostatic interactions. These assignments are supported by charges and chemical bonding analysis, where the interacting quantum atoms (IQA) method plays a fundamental role, identifying that central carbon sustains stabilizing interactions with the six surrounding atoms. However, our exploration of the potential energy surface reveals that these are not the global minimum structures. Nevertheless, the complexation of the $Ca_2C_5{}^{2+}$ system with two cyclopentadienyl anion ligands enhances the relative stability of the phC isomer. Moreover, when we harbor this structure within the cyclopentadienylene ([8]CPP) nano-ring, it provides protection and kinetic stability to the phC species. These results demonstrate that using suitable ligands is a promising strategy that deserves more systematic study in the search for new viable phC systems.

**Supplementary Materials:** The following are available online at https://www.mdpi.com/article/10.3390/atoms11030056/s1, Figures S1–S3: Adaptive Natural Density Partitioning bonding pattern of the $M_2C_5^{0/2+}$ (M = Be-Ca, Al-In, and Zn) at the ωb97XD/Def2-TZVP level, Figures S4–S9: Putative global minimum and low-lying isomers of $(C_6H_6)_2$-$M_2C_5$ (M = Li-K and Al-In), Figures S10–S13: Putative global minimum and low-lying isomers of $(C_5H_5^-)_2$-$M_2C_5^{2+}$ (M = Be-Ca and Zn), Figure S14: Putative global minimum and low-lying isomers of $[8\text{-CPP}]^{2-}[Ca_2C_5]^{2+}$, Table S1: Cartesian coordinates of the $M_2C_5^{0/2+}$ (M = Li-K, Be-Ca, Al-In, and Zn) phC minima optimized structures at the ωb97XD/def2-TZVP level and Table S2: Cartesian coordinates of the $[8\text{-CPP}]^{2-}[Ca_2C_5]^{2+}$ optimized structures at the ωb97XD/def2-TZVP level. Video S1: Video-Ca-CPP[8]. Video S2: Video-Ca-Cp2.

**Author Contributions:** Conceptualization, W.T., D.I. and O.Y.; methodology, D.I., O.Y., L.L.-P., J.S.-E. and A.V.-E.; software, D.I., O.Y., L.L.-P., J.S.-E. and A.V.-E.; validation, W.T., O.Y., D.I. and M.L.V.; formal analysis, W.T., O.Y. and D.I.; investigation, W.T., D.I. and O.Y.; resources, W.T. and M.L.V.; data curation, D.I., A.V.-E. and J.S.-E.; writing—original draft preparation, W.T., D.I. and O.Y.; writing—review and editing, all authors.; visualization, W.T., O.Y. and J.S.-E.; supervision, W.T. and M.L.V.; project administration, W.T.; funding acquisition, W.T. and M.L.V. All authors have read and agreed to the published version of the manuscript.

**Funding:** This research was funded by the National Agency for Research and Development (ANID) through FONDECYT projects 1211128 (W.T.) and 1221019 (A.V.-E.).

**Data Availability Statement:** The data presented in this study are available in the Supplementary Materials.

**Acknowledgments:** We thank the National Agency for Research and Development (ANID)/Scholarship Program/BECAS DOCTORADO NACIONAL/2019-21190427 (D.I.). National Agency for Research and Development (ANID)/Scholarship Program/BECAS DOCTORADO NACIONAL/2020-21201177 (L.L.-P.). Scholarship Program/BECAS DOCTORADO UNAB (J.S.-E) Powered@NLHPC: This research was partially supported by the supercomputing infrastructure of the NLHPC (ECM-02).

**Conflicts of Interest:** The authors declare no conflict of interest.

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
