# Peer review of "Searching for Systems with Planar Hexacoordinate Carbons"

_atoms, doi:10.3390/atoms11030056_

Round 1

Reviewer 1 Report

This manuscript proposes (1) to refine previous characterization of the bonding in planar C5 complexes stabilized by attendant metals, and (2) to advance our understanding of these systems by means of MD simulations and automated searches of the PES.  In my opinion, the first aim is overstated and should in any case be subordinate to the second aim.  It is only a change in emphasis, but I recommend that the paper be recast to focus on the MD simulations and PES search results.

Regarding the first aim, the methodologies of the electronic structure and MD calculations are well chosen.  However, the question of whether to call these systems tetracoordinate or hexacoordinate is not fundamentally important, as the underlying physics appears to be described consistently between this work and the previous work.

On line 55, I believe that the 2004 Merino paper at issue is JACS 126 16160 (but there's also Merino, JACS 126 15309).  This paper is actually missing from the references, and of course it should be cited in this sentence.

I agree with the present authors that "hexacoordinate" is more accurate, but the authors of Merino et al.  in characterizing these systems as "tetracoordinate" were fully aware that the metals contributed to the stability of the systems through ionic bonding.  They state "... the parental C52- skeleton binds to the alkaline and alkaline earth atoms by a highly ionic bond." What they considered carbon coordination was simply limited to the covalent interactions.

The IQA and AdNDP analyses carried out in the present manuscript are convincing, but I would dispute that they lead to a significant reassessment of the bonding from Merino's work.  In contrast, the MD simulations and PES search are new contributions; hence my recommendation that those results should be the focus of the paper overall.

Only one comment on the electronic structure calculations: it's likely that the singlet states are indeed the lowest energy, but comparison with other unsaturated carbon systems mandates confirming that the triplets are higher energy.  If this has been established in previous work, that would be sufficient, but those tests should be cited by the authors.

MINOR COMMENTS

- Please define abbreviations when first used (GM, NPA, etc.).

- "Best minima" and "best isomers" should be "lowest energy -- ."

- \delta should be included in the Table 2 header.

- The language surrounding Fig. 1 is confusing.  The authors should simply stipulate what the total charge is, based on the AdNDP analysis, rather than describing only the breakdown of contributions.  Identifying a \pi-bond is not what confirms the depiction of these systems as C5(2-) interacting with two metal cations -- it's the sum charge analysis that makes the case.  Similarly, the last sentence of that section ("AdNDP also recovers the non-bonding electrons on the counterions when corresponding") is completely unclear.  Is this conclusion not supported by more primitive methods such as Mulliken charge analysis?

- Fig. 2 caption: Needs rearranging to clarify what the second number represents:
"Relative energies are shown in kcal mol-1 at ωb97XD/Def2-TZVP (in bold) level, including zero-point energy (ZPE) corrections and their lowest harmonic vibrational frequency" should be "Relative energies including zero-point energy (ZPE) corrections are shown in kcal mol-1 at ωb97XD/Def2-TZVP (in bold) level, and their lowest harmonic vibrational frequency"

Author Response

Please find our response in the attached file.

Reviewer 2 Report

The manuscript is devoted to systems with planar hexacoordinate carbons. The search of such systems was carried out with the AUTOMATON program. The bonding was analyzed with the AdNDP method, Pyykkö’s single-bond radii, and many others; Interacting Quantum Atoms interaction energy and Coulombic and exchange-correlation energy components are calculated and compared for M2C5. 

It is well-structured and written, the systematic study is organized and supplemented by figures. In most cases, the computational results are compared with the published articles. The details of methods are discussed in Computational Details. The conclusions are supported by the results. I can recommend this manuscript for Atoms journal after minor changes listed below:

- Abstract should be rewritten in a formal way to clearly state the problem and results of this research, without a historical overview.

- The Introduction section is too concise with [3-7], [12-21], so on. On the other hand, paragraphs staring with line 55: In 2004, Merino et al. reported... contain no references. A more detailed overview of the field is required. For example, lines 180-181 "The use of ligands to protect is a common technique to provide stability and viability to 180 atomic clusters [19,61,62]." and similar are supposed to be in the Introduction section. 

- subsec 3.1 should be started from line 106. 

- "In this work, we have raised some questions regarding the systems mentioned in the previous paragraph." This paragraph is supposed to reference more formal structure of the manuscript, e.g., section 3, subsection 3.1, so on.

- Figure 4 is given to illustrate boxplot of RMSD fluctuation of ab initio BOMD simulations for [(C5H5–)]2[Ca2C5]2+ and [8- 211 CPP]2–[Ca2C5]2+ systems. However, the numerical data of this plot are not written in the text. 

After minor changes, the manuscript can be accepted to Atoms. 

Author Response

(The authors gave the same response as above.)
